# The Effect of Alkali Iodide Salts in the Inclusion Process of Phenolphthalein in β-Cyclodextrin: A Spectroscopic and Theoretical Study

**DOI:** 10.3390/molecules28031147

**Published:** 2023-01-23

**Authors:** Constantine Kouderis, Stefanos Tsigoias, Panagiota Siafarika, Angelos G. Kalampounias

**Affiliations:** 1Department of Chemistry, University of Ioannina, GR-45110 Ioannina, Greece; 2University Research Center of Ioannina (URCI), Institute of Materials Science and Computing, GR-45110 Ioannina, Greece

**Keywords:** β-cyclodextrin, phenolphthalein, inclusion complex, UV–Vis, FT-IR, molecular docking

## Abstract

The formation of the inclusion complex between β-cyclodextrin (CD) and phenolphthalein (PP) was investigated by means of UV–Vis and FT-IR spectroscopies. The thermodynamic parameters were calculated in the absence and presence of LiI, KI, NaI and CsI iodide salts. The enthalpy change during the formation was found to be negative for all solutions with iodide salts. The enthalpy change was found to decrease in the sequence no salt > NaI > KI> CsI > LiI. Moreover, it was observed that with increasing salt concentration enthalpy decreases monotonically. The interaction between the two molecules was mostly attributed to hydrogen bonding and Van der Waals interactions. Thermodynamic properties revealed that electrostatic forces also contribute when LiI is present in solutions. A molecular docking study was performed to elucidate the docking between phenolphthalein and cyclodextrin. The FT-IR spectra of CD, PP and the CD–PP complex were recorded to establish the formation of the inclusion complex. Semi-empirical and DFT methods were utilized to study theoretically the complexation process and calculate the IR vibrational spectra. The adequate agreement between theoretical and experimental results supports the proposed structural model for the CD–PP complexation.

## 1. Introduction

The interest in the inclusion complexes has arisen in the last forty years [1]. The reason behind this is the number of the applications that those complexes have. They can be used as model systems for the study of non-covalent interactions between molecules and in applications such as the encapsulation of sensitive photoactive molecules for preservation, drug delivery or even the filtration of water. The inclusion complexes consist of a host and a guest molecule. The cavity of the host molecule and the type of the interactions induced between host and guest molecule are the key factors for the encapsulation of the guest molecule. Cyclodextrins (CDs) are widely used as host molecules [1,2,3,4]. 

Cyclodextrins are cyclic oligomers of glucose units that are connected via *a*-(1,4)-bonds. Cyclodextrins can be categorized based on the possessed number of the glucose units involved in their structure. A-cyclodextrin has six, β-cyclodextrin has seven, and γ-cyclodextrin has eight *a*-glucopyranose rings. Cyclodextrins can be found in nature or be synthesized in the laboratory. They form a toroidal shape with a hydrophobic inner cavity and a hydrophilic outer surface [1,2,3,4]. Due to their characteristic molecular structure, CDs have the ability to act as host molecules for other smaller molecules, forming a host–guest complex through the formation of non-covalent bonds, with potential application in drug delivery, the food industry, water purification, etc. [5,6,7] The guest molecules can vary in polarity and ability to form hydrogen bonds. CDs can also be used for enhancing the solubility of a molecule in an aquatic environment, and improve bioavailability or release control [4,8]. Additional information concerning β-cyclodextrin and derivatives can be found in [9,10], while for β-cyclodextrin–guest interactions in [11,12,13].

Cyclodextrin inclusion complexes are influenced by the presence of inorganic salts. Salts are usually used to control pH of the aqueous solutions. Nevertheless, salts can also affect the complexation coefficient by altering the structure of water around cyclodextrin with subsequent variations in the hydrophobic interactions between host and guest molecules [2,14]. Anions and cations of inorganic salts can compete with the guest molecule for the binding sites of cyclodextrin [2]. The complexation process of β-cyclodextrin in an aqueous environment and how temperature and concentration affect this process have been the subject of various techniques including UV/Vis spectrometry for the estimation of the complexation coefficient, vibrational spectroscopy, NMR spectroscopy and others [15] (and references therein). 

In this work, we studied in detail the complexation of β-cyclodextrin with phenolphthalein as a function of concentration and temperature by means of spectrophotometric techniques and theoretical methods. The addition of inorganic salts with different-sized cations on the equilibrium constant was also examined. A molecular docking study was performed to gain information regarding the binding sites and the corresponding binding energy. The associated thermodynamic parameters, as well as the vibrational properties, were estimated by quantum mechanical theoretical calculations and compared with the related experimental values. The competition of hydrophobic molecules for the cyclodextrin cavity is well known and reported in the literature. However, it is not so trivial how inorganic salts, small cations and anions may affect the inclusion complex formation, and thereby the functional properties of CDs. This topic is recently of particular interest for formulators since there are several granted patents and published patent applications on commercially available CD derivatives with certain intrinsic ion contents. The experimental and theoretical approach by means of molecular docking and DFT methodologies presented in this work enable us to study the complexation in depth.

## 2. Materials and Methods

### 2.1. Solutions

β-cyclodextrin (Sigma-Aldrich, Burlington, MA, USA, purity 99%) and phenolphthalein (Merck KGaA, Darmstadt, Germany, purity 98%) were dissolved in triply distilled water. No further purification was attempted. 

Phenolphthalein (PP) has been employed for over a century as a laxative, while later it was found suspicious for carcinogenesis [16]. In a chemical laboratory, PP is usually used as a pH indicator as it can switch its color from colorless to fuchsia when pH changes from acidic to basic [17,18]. The concentration of phenolphthalein was kept constant at 5 × 10^−5^ M, while the concentration of β-cyclodextrin varied between 3.32226 × 10^−5^ M and 0.00125 M with a step of 3.2 × 10^−5^. A sodium carbonate solution was used to control pH near ~10 for phenolphthalein to present fuchsia color. The concentration of the sodium carbonate solution was 0.02 M. LiI (Alfa Aesar, Karlsruhe, Germany, purity: 99%), NaI (Alfa Aesar, purity: 99.5%), KI (Alfa Aesar, purity: 99%) and CsI (Alfa Aesar, purity: 99.99%) were added in the CD–PP solutions at the desired concentration. The molar concentrations of the prepared solutions are presented in Appendix A of the Appendix A. All measurements were performed in fresh solutions to avoid aging.

We used the so-called layering technique to receive the CD–PP complex in solid state from the solution. The layering technique, also known as the liquid diffusion technique, involves the slow diffusion of one solvent into another [19]. The two solvents were added to a test tube to form distinct layers. The solvent in the bottom layer was chosen because it can dissolve the compound satisfactorily. The solvent in the upper side was less dense than the solvent in the bottom and the compound should be insoluble in this solvent. The upper solvent started to diffuse in the bottom one, lowering the solubility of the latter. As the diffusion proceeded, solubility decreased further, forcing the compound to precipitate. Finally, solid crystals were collected with the aid of a filter. In this study, diethyl ether was the upper solvent and water was the solvent at the bottom. Smaller test tubes were used to achieve slower diffusion and thus receive large and well-shaped crystals. A schematic representation of the β-cyclodextrin and phenolphthalein is presented in Figure 1. 

### 2.2. Spectrophotometric Measurements

The UV-1600 PC series spectrophotometer by VWR (Radnor, PA, USA) was used to record electronic absorption spectra in the wavelength range from 190 to 1100 nm. The pathlength of the quartz cell was 1 cm. All the measurements were performed at 20 °C with an accuracy of ± 0.01 °C [20].

All infrared spectra in the 4000 to 370 cm^−1^ spectral region were obtained from KBr discs at room temperature with an Alpha spectrometer by Bruker. The spectral resolution for all measurements was set at 2 cm^–1^. The absorption of the KBr was applied as a blank to calibrate the spectrometer [21].

### 2.3. Theoretical Calculations

Molecular docking calculations are a useful tool for the estimation of the free binding energy of a complexation process. The energy is calculated using a semi-empirical force field to determine the best conformations. The force field has been parameterized using a library full of docking complexes with a known structure and complexation constants. The evaluation of the binding energy is a two-step process. The guest and host molecules start in an unbound conformation. In the first step, the intramolecular energetics are estimated for the transition from these unbound states to the conformation of the guest and host in the bound state. The second step then evaluates the intermolecular energetics of combining the guest and host in their bound conformation. More details concerning the molecular docking method can be found in [22].

The applied force field consist of six pair-wise terms (V) and a term related with the approximation of the entropy loss during binding [23]:(1)$$\Delta G=\left(V_{\mathrm{bound}}^{G-G}{-V}_{\mathrm{unbound}}^{G-G}\right)+\left(V_{\mathrm{bound}}^{H-H}{-V}_{\mathrm{unbound}}^{H-H}\right){+(V}_{\mathrm{bound}}^{H-G}{-V}_{\mathrm{unbound}}^{H-G}{+\Delta S}_{\mathrm{conf}})$$
where symbols G and F refer to guest and host molecules. Every pair-wise energy term includes assessments of hydrogen bonding, electrostatics, dispersion and desolvation [23]:(2)$${V=W}_{\mathrm{vdw}}\underset{i,j}{\sum }\left(\frac{A_{\mathrm{ij}}}{r_{\mathrm{ij}}^{12}}-\frac{B_{\mathrm{ij}}}{r_{\mathrm{ij}}^{6}}\right){+W}_{\mathrm{hbond}}\underset{i,j}{\sum }\left(\frac{C_{\mathrm{ij}}}{r_{\mathrm{ij}}^{12}}-\frac{D_{\mathrm{ij}}}{r_{\mathrm{ij}}^{10}}\right){+W}_{\mathrm{elec}}\underset{i,j}{\sum }\frac{q_{i}q_{j}}{{e(r}_{\mathrm{ij}}{)r}_{\mathrm{ij}}}{+W}_{\mathrm{sol}}\underset{\mathrm{ij}}{\sum }{(S}_{i}V_{j}{+S}_{j}V_{i}{)e}^{{(-r}_{\mathrm{ij}}^{2}/{2\sigma }^{2})}$$

Each term has been optimized by the weighting constants W. The weighting constant is used to adjust the empirical free energy that depends on a set of experimentally calculated binding constants. The first term stands for dispersion/repulsion with a 6/12 potential. The second term relates to hydrogen bonds with a 10/12 potential. The parameters C and D were appointed to give a max well depth for hydrogen bonds with oxygen and nitrogen. The well has a depth of 5 kcal/mol. The third term stands for Coulomb potential for electrostatics. The final term refers to the desolvation potential that entails from the volume of the atoms (V) that surround an atom and shield it from the solvent. The desolvation term is also weighted by a solvation parameter (S). There is also a term that includes the distance-weighting factor σ = 3.5 Å [23].

In the context of the Lamarckian genetic algorithm (LGA), all the possible ligand poses are generated and optimized repeatedly. The search for the best guest–host interaction is finalized when the number of evaluations reaches its limit. The LGA also uses an extra tool named *local search*. This tool runs a fixed number of iterations that account for the rotational, translational, and torsional degrees of freedom of the ligands. These ligands are randomly selected by each step after several successful and unsuccessful tests [24]. 

In this work, we utilized AutoDock 4.2 (Scripps Research, San Diego, CA, USA) for the molecular docking study. We implemented the Lamarckian Genetic Algorithm in collaboration with a grid-based energy-computation method for evaluating the grid maps. This method examines all the interatomic interactions of each atom present in the host and guest molecules. β-cyclodextrin was selected as the host and phenolphthalein as the guest molecule for the docking study. We used a grid box with dimensions of 40 × 40 × 40 Å and grid spacing of 0.375 Å. Phenolphthalein was free of restrains, while the opposite holds for cyclodextrin. In the guest molecule, four rotatable bonds were given by default. From all the runs, only ten of the poses were given in the output. The pose with the highest score was chosen as the most stable and was used in molecular orbital calculations.

We obtained the enthalpy of the host–guest complexation by means of the Gaussian 09 W Revision A.02 package of programs [25]. We also applied semi-empirical methods to determine the enthalpy. The optimization of the molecules was conducted using PM3 and AM1 in vacuo. The host and guest molecules, as well as the complex were optimized. The optimized structures of the host and guest molecules were used as input for the docking study. The enthalpy change was calculated as:(3)ΔH=∑νΔH(products)−∑νΔH(reactants)
where ν denotes the stoichiometric coefficients of the products and reactants.

## 3. Results and Discussion 

### 3.1. Calculation of the Complexation Coefficient

Figure 1 illustrates the electronic absorption spectra of the phenolphthalein/β-cyclodextrin solution. The addition of β-cyclodextrin leads to the continuous decrease of the peaks observed in the spectrum. The main peaks are located at 552.1, 375.2 and 292.7 nm and are attributed to phenolphthalein. The strongest peak is observed in the visible region of the spectrum. After the addition of β-cyclodextrin, the color of the solution changed from fuchsia to colorless indicating the formation of the inclusion complex.

The structure of PP alters when encapsulated in β-cyclodextrin and the color of the solution transforms from pink to colorless. The absorption of pure PP is:(4)$$A_{0}{=\varepsilon }_{552}{b[\mathrm{PP}]}_{0}$$
where b is the optical path length, ε_552_ is the molar attenuation coefficient or absorptivity of the attenuating species at the 552 nm wavelength and [PP]_0_ is the is the concentration of the pure PP.

The absorption of PP after the addition of CD is given by:(5)$${A=\varepsilon }_{552}b[\mathrm{PP}]$$ By dividing Equations (4) and (5), the ratio R_1_ is estimated as:(6)R1=A0A=[PP]0[PP] From the conservation of mass, we have:(7)$${\left[\mathrm{PP}\right]}_{0}=\left[\mathrm{PP}\right]+[\mathrm{CD}-\mathrm{PP}]$$
(8)$${\left[\mathrm{CD}\right]}_{0}=\left[\mathrm{CD}\right]+[\mathrm{CD}-\mathrm{PP}]$$ The complexation coefficient Kc is:(9)$$K_{C}=\frac{\left[\mathrm{CD}-\mathrm{PP}\right]}{\left[\mathrm{CD}\right]\left[\mathrm{PP}\right]}$$ By combining the above equations, we obtain the following formula for the complexation coefficient:(10)$$K_{C}=\frac{{-\;(1-R}_{1})}{R_{1}{\left[\mathrm{CD}\right]}_{0}+{\left[\mathrm{PP}\right]}_{0}{(1-R}_{1})}$$
(11)$$R_{1}=1/2(1+\left({\left[\mathrm{CD}\right]}_{0}-{\left[\mathrm{PP}\right]}_{0}\right){)K}_{C}+\sqrt{{\left[\mathrm{CD}\right]}_{0}^{2}K_{C}^{2}-2{\left[\mathrm{CD}\right]}_{0}K_{C}\left({-1+K}_{C}{\left[\mathrm{PP}\right]}_{0}\right)+{\left({1+K}_{C}{\left[\mathrm{PP}\right]}_{0}\right)}^{2}}$$
where A_0_ is the absorption of phenolphthalein solution before the addition of β-cyclodextrin and A is the absorption of the solution after the addition of β-cyclodextrin. The concentration of cyclodextrin is denoted as [CD], while K _C_ is the complexation coefficient. In the latter equation, the complexation coefficient is the only unknown parameter. 

Figure 2a demonstrates the absorbance of the CD–PP solution in the presence of different alkali iodide salts corresponding to the same salt concentration of 100 mM. An emerged variation is observed in absorbance with changing the alkali type. As cation size increases, the absorbance rises except for Li, which has the smallest ionic radius. Figure 2b shows the solution absorbance as a function of wavelength for several KI concentrations. It seems that the maximum absorbance increases gradually with salt concentration for a specific alkali-type.

The variation of the A_0_/A ratio as a function of β-cyclodextrin concentration after the addition of different alkali iodide salts is presented in Figure 3a. From the plot of the ratio R_1_ = A_0_/A versus β-cyclodextrin concentration, the complexation coefficient Kc can be estimated by fitting the experimental results with Equation (11). The results indicate that the complexation coefficient is reduced after the addition of iodide salts due to the presence of the I anion which acts as a competitor for the complexation. The complexation coefficient variation is not monotonically related with the ionic radius of the alkali. This behavior can be explained having in mind that inorganic salts change the activity of the free and complexed phenolphthalein. Salts may also change the activity of the cyclodextrin. Nevertheless, this effect is negligible due to the neutral character of the β-cyclodextrin molecule. Furthermore, salts may compete with phenolphthalein for the complexation sites of β-cyclodextrin [2,16,26]. 

Figure 3b demonstrates the concentration dependence of the A_0_/A ratio for several KI concentrations. The increase in the KI salt concentration induces a monotonous reduction in the K_C_ values, which is expected since the I anion is a competitor for the complexation sites [26].

### 3.2. Thermodynamic Analysis

The Van’t Hoff equation relates the thermodynamic parameters ΔH, ΔS and ΔG associated with the complexation process with the complexation coefficient Kc, as: (12)$${\mathrm{lnK}}_{C}=-\frac{\Delta H}{\mathrm{RT}}+\frac{\Delta S}{R}$$
where R is the gas constant. Both ΔH and ΔS are assumed as temperature-independent. From the measured absorbance as a function of temperature, we estimated the corresponding complexation coefficients. In Figure 4 a,b are shown the plots of lnK_C_ as a function of reciprocal temperature 1/T for different alkali-types and salt concentrations, respectively. By applying Equation (12), the relevant thermodynamic parameters ΔS and ΔH were estimated from the slope and the intercept of the straight lines. The almost perfect linear dependency reveals that both ΔH and ΔS are temperature-independent in the range studied here. The Gibbs free energy change associated with the inclusion process was calculated for each temperature studied using the equation:(13)$${\Delta G=-\mathrm{RTlnK}}_{C}$$

The calculated values of the complexation coefficient and the thermodynamic parameters ΔH, ΔG and ΔS are summarized in Table 1. 

When phenolphthalein starts to bind in cyclodextrin, four types of interactions occur. These interactions are hydrogen bonding, van der Waals interactions, and hydrophobic and electrostatic forces [27]. From the sign of the thermodynamic parameters associated with the complexation process, the interplay between the interactions dominating the complexation process can be assessed. When ΔH > 0 and ΔS > 0, hydrophobic interactions dominate, while when ΔH < 0 and ΔS < 0, hydrogen bonding and van der Waals interactions play a major role. Electrostatic forces are considerable when the relations ΔH < 0 and ΔS > 0 hold [28]. The thermodynamic parameters presented in Table 1 are all negative, indicating that hydrogen bonding and van der Waals interactions are important in the studied system. The Gibbs free energy change overall decreases with increasing temperature, indicating that the process becomes less spontaneous and shifts the equilibrium towards the reactants [29].

A host–guest complex of CD is driven by several forces: Van der Waals interactions, hydrogen bonding, hydrophobic interactions, electrostatic forces and the release of water molecules from the cavity of the host to the bulk [30]. The value of the thermodynamic parameters is a contribution of the desolvation and the non-covalent interactions that we mentioned above. Generally, for the complexation of CDs the usual driving forces are hydrophobic interactions, van der Waals interactions and hydrogen bonding. Molecular docking results showed that van der Waals and hydrogen bonding predominate. When ΔH < 0 and ΔS < 0, the van der Waals interactions and hydrogen bonding are the driving forces for the complexation. The reason is that van der Waals interactions and hydrogen bonding between semi-polar molecules is an exothermic process. Moreover, the conformational restriction and loss of rotational freedom during the formation of the inclusion complex yields unfavorable entropy loss [31]. When ΔH > 0 and ΔS > 0, the hydrophobic interactions are involved in the complexation. The hydrophobic interactions are entropy- driven with a positive entropy change and a slightly positive enthalpy change. Lastly, when ΔH < 0 and ΔS > 0, the electrostatic forces play a significant role [32].

The complexation coefficient and the thermodynamic parameters ΔH, ΔG and ΔS for the β-cyclodextrin-phenolphthalein complex formation in the presence of salts are presented in Table 2. With increasing the concentration of the KI salt, the absolute value of the enthalpy change decreases due to the competition between iodine anions of KI and phenolphthalein for the binding sites of β-cyclodextrin [26,33]. 

The formation of the inclusion complex is a two-step procedure. Firstly, water from the cavity of cyclodextrin is released to the bulk and then the guest molecule is inserted into the cavity. In the presence of a competitor anion in the solution, the guest molecule should replace both water and the relatively strongly attached anion to cyclodextrin. Further, the heat of the PP–CD formation is the sum of the enthalpy of the expelling anion from the CD cavity and the CD–PP binding enthalpy. Some of the heat from the formation of the inclusion complex of CD–PP is used for the expulsion of the anion and thus, reduces the enthalpy value of the complexation. Furthermore, it seems that with increasing the alkali ionic radius of the cation in going from Na to Cs, the absolute value of ΔH decreases. In this case, Li does not follow this trend.

To explain this behavior, we should assess the role of the cations in the complexation process. The cations interact with both cyclodextrin and phenolphthalein molecules. In the present case, the guest molecule has one hydroxyl group and two oxygen atoms negatively charged in an extensive conjugative system. The cations may interact with these sites and therefore, affects the activity of phenolphthalein [34]. More specifically, the Li cation is a small density, positively charged ion producing a much larger effect than the other alkali ions. These interactions are reflected in the entropy change, as can be seen in Table 2. For all cations except Li, a relatively strong entropy change is shown. For Li, ΔS is almost zero indicating that electrostatic interactions play a key role in the complexation process. As a result, Li affects the formation of hydrogen bonds between CD and phenolphthalein [28,33]. The same effect is observed in the case of cyclodextrin, but to a lesser extent due to the neutral character of the cyclodextrin molecule [35]. 

Another way to influence the complexation process is through the hydration of the alkali metals. In an inclusion reaction, numerous water molecules participate. First, the structure of water and cyclodextrin must break down. Then, the CD–PP complex is formed, and water molecules are rearranged around the complex. When salt is added to the system, this process is disrupted by the cations through their hydration process. This phenomenon is more energy-consuming as the radius of the ion increases. Finally, cyclodextrin can form complexes with cations, although these complexes are very unstable [35]. 

The complexation of cyclodextrin with phenolphthalein as a function of pH has been studied in the past [36]. Only in one case, the pH conditions were the same as ours (pH = 10). Furthermore, in that study there was an excess concentration of cyclodextrin in solutions. Nevertheless, the outcome was the formation of a stable 1:1 CD–PP complex, which agrees with our results. The complexation constant of cyclodextrin with phenolphthalein in various temperatures was also estimated previously [37]. The complexation constants reported in that work are slightly higher than our results, although close enough. To the best of our knowledge, how inorganic salts may affect the CD–PP inclusion complex formation, and thereby the functional properties of CDs, is presented for the first time.

### 3.3. Spectroscopic Evidence of the Complex Formation

The IR absorption spectra of pure β-cyclodextrin, phenolphthalein and their complex were recorded in their crystalline form under ambient conditions. The analysis of the FT-IR spectra was limited in the 500–1600 cm^–1^ spectral region in an effort to confirm the formation of the inclusion complex. Figure 5a shows the vibrational spectra of the host, guest and complex molecules. 

Figure 5b depicts the CD and CD–PP complex spectra for direct comparison. The spectrum of the CD–PP complex resemble more the β-cyclodextrin spectrum than that of phenolphthalein. This is more or less expected; nevertheless, several spectral variations are observed due to the inclusion of phenolphthalein into β-cyclodextrin. A new weak band is observed near ~1240 cm^−1^. This peak is attributed to the characteristic vibration of the rings of the phenolphthalein. In pure phenolphthalein, this band is detected near ~1238 cm^–1^. The peak at ~1260 cm^–^1 is assigned to the aromatic ring vibration. This ring is directly connected with the pentagonal ring including the ester group of PP. In pure PP, the corresponding peak is observed at ~1254 cm^–1^. Another spectral feature associated with the inclusion process is the spectral envelope that is observed near ~1737 cm^–1^. This feature is attributed to the vibration of the carbonyl bond in the ester group of the phenolphthalein. In the low-frequency region, the spectrum of the complex exhibits a doublet at ~530 cm^–1^, present also in the spectrum of β-cyclodextrin, however with a difference relative intensity ratio. These spectral changes solidify the formation of the inclusion complex.

### 3.4. Molecular Docking and DFT Results 

For the theoretical study of the complexation, we implemented molecular docking and DFT calculations. Molecular docking allowed us to establish the docking between phenolphthalein and β-cyclodextrin molecules. In addition, DFT calculations were performed to optimize the structures of the molecules that were used as input in the docking study and to estimate theoretically the relevant thermodynamic parameters and the vibrational spectra. The molecular docking results are presented in Figure 6. It is seen that the PP molecule interacts with β-cyclodextrin through hydrogen bonding and Van der Waals forces. The corresponding binding energy was found to be equal to −7.20 kcal/mol, exhibiting a strong affinity between the two compounds.

All the alkali cations and iodine anions of the iodide salts were attempted to be docked in β-cyclodextrin without the presence of phenolphthalein. The interaction between each cation and CD was rather weak. Li and Na gave positive binding energy, indicating that a stable inclusion complex cannot be formed. K and Cs showed a slightly negative binding energy, leading to the formation of an unstable inclusion complex. The binding energy followed a monotonous decrease on going from lithium to cesium. However, the iodine anion exhibited a relatively strong interaction with β-cyclodextrin, giving a binding energy of −1.12 kcal/mol. 

The binding energies were used only qualitatively to elucidate which ligand participates in the complexation process. The greater binding energy corresponding to PP was found almost two orders of magnitude higher than the lowest energy corresponding to cations. The errors that usually appear in molecular docking studies lie between 2–3 kcal/mol as reported in the literature [38,39,40]. It is true that the binding energies may not be very accurate to be used quantitatively; therefore, we did not proceed to further quantitative estimations using this methodology. The iodine anion and the alkali cations may form inclusion complexes with cyclodextrins, but the anions usually form stronger complexes relative to cations [35,41]. When a salt is added in the solution and we have to compare the anion and the cation in terms of their ability to form inclusion complexes, in most of the cases the anion has a greater ability [42].

The AM1 and PM3 semi-empirical methods were applied for the estimation of the enthalpy change of the complexation process with the latter exhibiting higher accuracy. The enthalpy change was calculated as −13.285 kcal/mol by PM3 and −6.123 kcal/mol by AM1. The latter value is close to the experimental value of −9.82 kcal/mol. 

The theoretical IR vibrational spectrum of the inclusion complex was calculated by the B3LYP method combined with the 6–311 g basis set. A direct comparison between the theoretical and the experimental spectrum is presented in Figure 7. Both spectra reveal a close resemblance at least in the number of peaks, supporting the proposed complexation between CD and PP molecules. The observed differences in the relative intensities are probably due to the fact that the vibrational frequencies for each normal mode were calculated in the gas phase without adjusting force constants. No imaginary vibrational frequencies were found in the theoretical calculation. 

## 4. Conclusions

UV–Vis spectrometry, FT-IR spectroscopy and theoretical calculations were combined to study the complexation process between phenolphthalein and β-cyclodextrin molecules in aqueous solutions and to determine the relevant thermodynamic properties. The effect of the addition of alkali salts in the complexation process was also investigated in detail. 

With increasing the concentration of the KI salt, the absolute value of the enthalpy change decreases due to the competition between iodine anion and phenolphthalein for the binding sites of β-cyclodextrin. Moreover, with increasing the alkali ionic radius of the cation on going from Na to Cs, the absolute value of ΔH decreases. Nevertheless, Li does not follow this trend. Hydrogen bonding and Van der Waals interactions dominate the complexation process, while in the case of lithium electrostatic forces are also important. As explained earlier, the sign of the ΔH and ΔS values allows us to elucidate in a qualitative manner the type of the interactions that dominate in the complexation. The experimental results revealed that the CD–PP complexation process in the presence of Li+ exhibits a slightly negative ΔS value, indicating that the electrostatic interactions play a significant role in the complexation compared to the other cations. Nevertheless, since the value of ΔS is slightly negative, it seems that van der Waals and hydrogen bonding still contribute to the formation of the complex.

The formation of the CD–PP complex was established by molecular docking calculations revealing hydrogen bonding and Van der Waals interactions between phenolphthalein and β-cyclodextrin molecules. DFT calculations were used to assess theoretically the thermodynamic parameters associated with the complexation process and predict the corresponding vibrational spectra. Both theoretical and experimental spectra revealed a close resemblance, supporting the proposed complexation mechanism between CD and PP molecules. 

## Data Availability

Data are available upon request from the corresponding author.

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
