# Peer review of "The Effect of Alkali Iodide Salts in the Inclusion Process of Phenolphthalein in β-Cyclodextrin: A Spectroscopic and Theoretical Study"

_molecules, 2023, doi:10.3390/molecules28031147_

Round 1

Reviewer 1 Report

This paper presents the effects of alkali metal salts on the complexation between β-CD and phenolphthalein. I want to make several critical comments.

-It’s better to move Table 1 to Supp. Info., because the molar concentrations of the solutions are not relevant to discussions in the main text.

-Using the binding energies for complexation processes is a very risky business, because the calculations for these parameters are usually not accurate (BSSE etc.). The authors state that (Line 370-374):

The interaction between each cation and CD was rather weak. Li and Na gave positive binding energy, indicating that a stable inclusion complex cannot be formed. K and Cs showed a slightly negative binding energy, leading to the formation of an unstable inclusion complex.

This is hard to believe, because β-CD possesses numerous electronegative O atoms. Are the authors saying that only I- binds to the β-CD, while Li+ or Na+ remains in solution, separated from β-CD? Moreover, among the cations Li+ should form the strongest interactions with β-CD because of the smallest size (and largest positive charge density). I recommend the authors to carry out DFT calculations for the MI - β-CD (M = Li, Na, K, Cs) complexes to see whether there exist stable complexes. My guess is that M+ interacts with the O atoms, whereas I- with the hydroxyl H atoms.

-The authors state that (Line 280-285):

From the sign of the thermodynamic parameters associated with the complexation process, the interplay between the interactions dominating the complexation process can be assessed. When ΔH > 0 and ΔS < 0, hydrophobic interactions dominate, while when ΔH < 0 and ΔS < 0, hydrogen bonding and van der Waals interactions play a major role. Electrostatic forces are considerable when the relations ΔH < 0 and ΔS > 0 hold [21].

Although the authors cite a paper, I do not understand this. The authors should give the rationale for this claim in more detail.

-The English expressions in the following should be revised:

Line 89-90: The solvent in the bottom side was chosen so that the 89 compound to be moderately soluble.

Line 317: The same effect has in cyclodextrin

-I do not understand why transition state is involved in the following paragraph (Line 118-119):

In this step, the intramolecular energies are calculated for the transition state and then for the conformations of the complex in these unbound states.

-Most readers (including me) may not be familiar with the Molecular docking method. The authors may cite some references for this.

-The errors in citing Ref. 2 and 3 should be corrected:

2. Contributions of weak interactions to the inclusion complexation of 3-hydroxynaphthalene-2-carboxylic acid and its analogues with cyclodextrins. Zheng-Ping Yi, Hui-Lan Chen, Zheng-Zi Huang, Qing Huang, Jun-Shen Yu. J. Chem. Soc., Perkin trans., 2000, 2359 to 2050. 432

3. A. Fifere, N. Marangoci, S. Maier, A. Coroaba, D. Maftei and M. Pinteala. Theoretical study on β-cyclodextrin inclusion complexes with propiconazole and protonated propiconazole. Theoretical study on β-cyclodextrin inclusion complexes with propiconazole and protonated propiconazole. J. Org. Chem, 2021, 8, P2191 – 2201.

-Citation of References is not good. Please cite the following related papers on β-cyclodextrin and derivatives:

Crini, G.; Review: A History of Cyclodextrins, Chem. Rev. 2014114, 10940-10975.

Lee, J.-u., Lee, S-S., Lee, S., Oh, H. B., Noncovalent Complexes of Cyclodextrin with Small

Organic Molecules: Applications and Insights into Host–Guest Interactions in the Gas Phase and Condensed Phase, Molecules 2020, 25, 4048.

-And for β-cyclodextrin – guest interactions:

Choi, H., Oh, Y.‑H., Park, S., Lee, S.‑S.; Oh, H. B., Lee, S. Unveiling host–guest–solvent interactions in solution by identifying highly unstable host–guest configurations in thermal non‑equilibrium gas phase, Sci. Rep. 2022, 12, 8169.

Hu, Q.-D., Tang, G.-P. & Chu, P. K. Cyclodextrin-based host–guest supramolecular nanoparticles for delivery: from design to applications. Acc. Chem. Res. 2014, 47, 2017–2025.

Zhan, W., Wei, T., Yu, Q. & Chen, H. Fabrication of supramolecular bioactive surfaces via β-cyclodextrin-based host–guest interactions. ACS Appl. Mater. Interfaces 2018, 10, 36585–36601.

Author Response

  • Reviewer #1 comments:

This paper presents the effects of alkali metal salts on the complexation between β-CD and phenolphthalein. I want to make several critical comments.

Issue 1:

It’s better to move Table 1 to Supp. Info., because the molar concentrations of the solutions are not relevant to discussions in the main text.

Reply to Reviewer comment and changes made:

We agree with reviewer, and we moved Table 1 into Supporting Information. All Tables of the revised manuscript were renumbered, accordingly.

Issue 2:

Using the binding energies for complexation processes is a very risky business, because the calculations for these parameters are usually not accurate (BSSE etc.). The authors state that (Line 370-374):

 The interaction between each cation and CD was rather weak. Li and Na gave positive binding energy, indicating that a stable inclusion complex cannot be formed. K and Cs showed a slightly negative binding energy, leading to the formation of an unstable inclusion complex.

 This is hard to believe, because β-CD possesses numerous electronegative O atoms. Are the authors saying that only I- binds to the β-CD, while Li+ or Na+ remains in solution, separated from β-CD? Moreover, among the cations Li+ should form the strongest interactions with β-CD because of the smallest size (and largest positive charge density). I recommend the authors to carry out DFT calculations for the MI - β-CD (M = Li, Na, K, Cs) complexes to see whether there exist stable complexes. My guess is that M+ interacts with the O atoms, whereas I- with the hydroxyl H atoms.

Reply to Reviewer comment and changes made:

The binding energies were used only qualitatively to elucidate which ligand participates in the complexation process. The greater binding energy corresponding to PP was found almost two orders of magnitude higher than the lowest energy corresponding to cations. The errors that usually appear in molecular docking studies lye between 2-3 Kcal/mol as reported in the literature:

doi.org/10.1021/acs.jcim.9b00778

doi.org/10.3390/polym14173690

doi.org/10.3390/molecules22111801

It is true that the binding energies may not be very accurate to be used quantitatively, therefore we did not proceed to further quantitative estimations using this methodology.

The iodine anion and the alkali cations may form inclusion complexes with cyclodextrins, but the anions usually form stronger complexes relative to cations

doi.org/10.1023/A:1021242310024

doi.org/10.1080/10610279308029828

When a salt is added in the solution and we have to compare the anion and the cation in terms of their ability to form inclusion complexes, in most of the cases anion has a greater ability [doi.org/10.1007/BF01076992]

These comments and six new references were added in the revised manuscript (lines 418-428). The reference list is renumbered, accordingly. 

Issue 3:

The authors state that (Line 280-285):

From the sign of the thermodynamic parameters associated with the complexation process, the interplay between the interactions dominating the complexation process can be assessed. When ΔH > 0 and ΔS < 0, hydrophobic interactions dominate, while when ΔH < 0 and ΔS < 0, hydrogen bonding and van der Waals interactions play a major role. Electrostatic forces are considerable when the relations ΔH < 0 and ΔS > 0 hold [21].

 Although the authors cite a paper, I do not understand this. The authors should give the rationale for this claim in more detail.

Reply to Reviewer comment and changes made:

A host-guest complex of CD has several forces which drive it. Van der Waals interactions, hydrogen bonding, hydrophobic interactions, electrostatic forces, and the release of water molecules from the cavity of the host to the bulk [doi.org/10.1021/cr960371r]. The value of the thermodynamic parameters is a contribution of the desolvation and the non-covalent interactions that we mentioned above. Generally, for the complexation of CDs the usual driving forces are hydrophobic interactions, van der Waals interactions and hydrogen bonding. Molecular docking results showed that van der Waals and hydrogen bonding predominate. When ΔH<0 and ΔS<0, the van der Waals interactions and hydrogen bonding are the driving forces for the complexation. The reason is that van der Waals interactions and hydrogen bonding between semi polar molecules is an exothermic process. Moreover, the conformational restriction and loss of rotational freedom during the formation of the inclusion complex yields unfavorable entropy loss [doi.org/10.1039/C9NR05790K]. When ΔΗ>0 and ΔS>0, the hydrophobic interactions are involved in the complexation. The hydrophobic interactions are entropy driven with a positive entropy change and a slightly positive enthalpy change. Lastly, when ΔΗ<0 and ΔS>0 the electrostatic forces play a significant role [doi.org/10.3390/molecules24244565].

These comments were added in the revised manuscript as a new paragraph and highlighted with yellow color. Three new references were also added, and the reference list is renumbered, accordingly (lines: 298-312).

Issue 4:

The English expressions in the following should be revised:

 Line 89-90: The solvent in the bottom side was chosen so that the 89 compound to be moderately soluble.

 Line 317: The same effect has in cyclodextrin

Reply to Reviewer comment and changes made:

We revised English language in several part of the manuscript.

The expression is replaced by: “The solvent in the bottom layer was chosen because it can dissolve the compound satisfactorily” (Lines: 95-96)

The expression is replaced by: “A similar effect is observed in the case of cyclodextrin,..” (Line 348)

Issue 5:

I do not understand why transition state is involved in the following paragraph (Line 118-119):

In this step, the intramolecular energies are calculated for the transition state and then for the conformations of the complex in these unbound states.

Reply to Reviewer comment and changes made:

We rewritten this part in order to avoid any misunderstanding. In general, the energy is calculated using a semi-empirical force field to determine the best conformations. The force field has been parameterized using a library full of docking complexes with a known structure and complexation constants. The evaluation of the binding energy is a two-step process. The guest and host molecules start in an unbound conformation. In the first step, the intramolecular energetics are estimated for the transition from these unbound states to the conformation of the guest and host in the bound state. The second step then evaluates the intermolecular energetics of combining the guest and host in their bound conformation. The addition in the revised manuscript is highlighted with yellow color (lines: 123-127).

Issue 6:

Most readers (including me) may not be familiar with the Molecular docking method. The authors may cite some references for this.

Reply to Reviewer comment and changes made:

We added two new references [doi.org/10.1002/jcc.21256 and https://doi.org/10.1002/jcc.21256] for more details concerning the molecular docking method (lines: 127-128). The reference list is renumbered, accordingly.

Issue 7:

The errors in citing Ref. 2 and 3 should be corrected:

  1. Contributions of weak interactions to the inclusion complexation of 3-hydroxynaphthalene-2-carboxylic acid and its analogues with cyclodextrins. Zheng-Ping Yi, Hui-Lan Chen, Zheng-Zi Huang, Qing Huang, Jun-Shen Yu. J. Chem. Soc., Perkin trans., 2000, 2359 to 2050. 432
  2. A. Fifere, N. Marangoci, S. Maier, A. Coroaba, D. Maftei and M. Pinteala. Theoretical study on β-cyclodextrin inclusion complexes with propiconazole and protonated propiconazole. Theoretical study on β-cyclodextrin inclusion complexes with propiconazole and protonated propiconazole. J. Org. Chem, 2021, 8, P2191 – 2201.

Reply to Reviewer comment and changes made:

We corrected these references according to reviewers’ comments.

Issue 8:

Citation of References is not good. Please cite the following related papers on β-cyclodextrin and derivatives:

 Crini, G.; Review: A History of Cyclodextrins, Chem. Rev. 2014, 114, 10940-10975.

 Lee, J.-u., Lee, S-S., Lee, S., Oh, H. B., Noncovalent Complexes of Cyclodextrin with Small

Organic Molecules: Applications and Insights into Host–Guest Interactions in the Gas Phase and Condensed Phase, Molecules 2020, 25, 4048.

Reply to Reviewer comment and changes made:

We added the suggested references concerning β-cyclodextrin and derivatives in the revised manuscript and the reference list is renumbered, accordingly. (lines: 48-49)

Issue 9:

And for β-cyclodextrin – guest interactions:

 Choi, H., Oh, Y. H., Park, S., Lee, S. S.; Oh, H. B., Lee, S. Unveiling host–guest–solvent interactions in solution by identifying highly unstable host–guest configurations in thermal non equilibrium gas phase, Sci. Rep. 2022, 12, 8169.

 Hu, Q.-D., Tang, G.-P. & Chu, P. K. Cyclodextrin-based host–guest supramolecular nanoparticles for delivery: from design to applications. Acc. Chem. Res. 2014, 47, 2017–2025.

 Zhan, W., Wei, T., Yu, Q. & Chen, H. Fabrication of supramolecular bioactive surfaces via β-cyclodextrin-based host–guest interactions. ACS Appl. Mater. Interfaces 2018, 10, 36585–36601.

Reply to Reviewer comment and changes made:

We added the suggested references concerning β-cyclodextrin – guest interactions in the revised manuscript and the reference list is renumbered, accordingly. (lines: 48-49)

Reviewer 2 Report

In this work the formation of the inclusion complex between β-cyclodextrin (CD) and phenolphthalein (PP) has been investigated using UV-Vis and FT-IR spectroscopies together with molecular docking and DFT methods. The obtained results might be of some potential interest, leading to some novel insights on the effect of alkyl iodide salts (LiI, NaI, KI, CsI) on the formation of the complex.

Nevertheless, the manuscript (MS) needs to be significantly improved and several relevant issues must be addressed in connection with data analysis, interpretation, and overall discussion. In my opinion, the manuscript cannot be published in the present form. It should be reconsidered after major revisions, clarifications, and improvements, as specified below.

 Major revisions

In my opinion, it should be better clarified why phenolphthalein (PP) has been considered in the present investigation. More importantly, since several studies dealing the complex formation between CD and PP already exist (see for instance: doi.org/10.1039/P29880001687 and doi.org/10.1016/S0731-7085(98)00150-2 and references therein), the authors must discuss their results also in connection with the previous findings, when obtained in similar conditions. By the way, some of the quantitative results (equilibrium constants) reported in the manuscript seem to differ considerably from the previous findings, obtained with similar methods (doi.org/10.1016/S0731-7085(98)00150-2). Overall, it is difficult to judge the relevance and suitability of the results provided by the present MS without any discussion with literature studies performed on the same complex.

P. 4 line 94: “In this study, water was the upper solvent and diethyl ether was the solvent at the bottom.” Likely the reverse is true. Please check.

P. 7. The trend reported in figure 3 are not consistent with those expected from Figure 2a and with what stated in line 226: “As cation size increases, the absorbance rises except for Li that has the smallest ionic radius.” Please check.

P.8 line 249: “The increase of the KI salt concentration induces a monotonous reduction of the KC values, which is expected since I anion is a competitor for the complexation” This can be also considered to explain the results mentioned above in line 235: “The results indicate that the complexation coefficient reduces after the addition of iodide salts.”, when different possibilities have been proposed (lines 237-242). Please clarify. I remark that if activity changes are considered, then equations (9) and (10) should be reformulated.

 Concerning the values reported in Table 2, I suggest expressing errors with only one meaningful digit; moreover the numbers of the related quantities must be expressed with an appropriate number of significant figures. In this respect it seems that the errors on DG are very high, also if compared to those on Kc. I remark that DG can be obtained directly from Kc, without the employ of eq. 13 that depends on DH and DS, which are, in turn, determined by the van’t Hoff plot.

P. 9 line 282: “When ΔH > 0 and ΔS < 0, hydrophobic interactions dominate, while when ΔH < 0 and ΔS < 0, hydrogen bonding and van der Waals interactions play a major role. Electrostatic forces are considerable when the relations ΔH < 0 and ΔS > 0 hold [21].” This description is very qualitative and difficult to be fully understood. For example, hydrophobic interactions are typically entropically driven. Please check.

P.9 line 287: “The Gibbs free energy change overall reduces with increasing temperature indicating that the process becomes less spontaneous [22]” The process is in any case at equilibrium, is the position of the equilibrium that changes with temperature, as expected for an exothermic process.  

P. 9 line 291:With increasing the concentration of the KI salt, the absolute value of the enthalpy change decreases due to the competition between iodine anion of KI and phenolphthalein for the binding sites of β-cyclodextrin [19, 23].” Why does the competition is expected to reduce the enthalpy value?

 P.10 lines 307-326. This part is quite assertive invoking different mechanisms and possibilities, but it is not clear how the findings of the present work might clarify this complex situation. Moreover, since iodide is expected to compete with PP, it is crucial to know if the formation of cation-anion ion pairs can be neglected in the considered conditions.

P.12 line 383: “Both spectra reveal a close resemblance at least in the number of peaks supporting the proposed complexation between CD and PP molecules.” It is very hard to understand how the comparison in Figure 7 might support the existence of the complex. Probably a comparison between the spectrum calculated for the free PP and that calculated for PP in the complex might be helpful to support the discussion reported in section 3.3, which in the present form is not very convincing. 

P.12 line 403: “Hydrogen bonding and Van der Waals interactions dominate the complexation process, while in the case of lithium electrostatic forces are also important.” This conclusion is not justified.

Minor revisions

Table 1 is very large; it is probably better to include it as supplementary material.

P. 4 line 87: “The layering technique, also known as the liquid diffusion technique, involves the slow diffusion of one solvent into another.” Probably one or more references should be cited here.

Author Response

  • Reviewer #2 comments:

In this work the formation of the inclusion complex between β-cyclodextrin (CD) and phenolphthalein (PP) has been investigated using UV-Vis and FT-IR spectroscopies together with molecular docking and DFT methods. The obtained results might be of some potential interest, leading to some novel insights on the effect of alkyl iodide salts (LiI, NaI, KI, CsI) on the formation of the complex.

Nevertheless, the manuscript (MS) needs to be significantly improved and several relevant issues must be addressed in connection with data analysis, interpretation, and overall discussion. In my opinion, the manuscript cannot be published in the present form. It should be reconsidered after major revisions, clarifications, and improvements, as specified below.

Issue 1:

In my opinion, it should be better clarified why phenolphthalein (PP) has been considered in the present investigation. More importantly, since several studies dealing the complex formation between CD and PP already exist (see for instance: doi.org/10.1039/P29880001687 and doi.org/10.1016/S0731-7085(98)00150-2 and references therein), the authors must discuss their results also in connection with the previous findings, when obtained in similar conditions. By the way, some of the quantitative results (equilibrium constants) reported in the manuscript seem to differ considerably from the previous findings, obtained with similar methods (doi.org/10.1016/S0731-7085(98)00150-2). Overall, it is difficult to judge the relevance and suitability of the results provided by the present MS without any discussion with literature studies performed on the same complex.

Reply to Reviewer comment and changes made:

The competition of hydrophobic molecules for the cyclodextrin cavity is well known and reported in the literature. However, it is not so trivial, how inorganic salts, small cations and anions may affect the inclusion complex formation, and thereby the functional properties of CDs. This topic is recently of particular interest for formulators since there are several granted patents and published patent applications on commercially available CD derivatives with certain intrinsic ion contents. The experimental and theoretical approach by means of molecular docking and DFT methodologies presented in this work, enable us to study the complexation in depth. We added these notes in the last paragraph of the Introduction section (lines: 67-74) in an effort to point out why we studied the effect of the addition of inorganic salts on the CD-PP complex formation.

We agree with reviewer’s comment that our results should be compared with the results of analogous studies on the same inclusion complex.

In the first suggested citation (doi.org/10.1039/P29880001687), the researchers have studied the complexation of cyclodextrin with phenolphthalein as a function of pH. Only in one case, the pH conditions were the same as ours (pH=10). Furthermore, in that study there was an excess concentration of cyclodextrin in their solutions and they used a mixed solvent (ethanol-water) instead of pure water as in our case. Nevertheless, the outcome of this study is the formation of a stable 1:1 CD-PP complex which is in agreement with our results.

In the second suggested citation (doi.org/10.1016/S0731-7085(98)00150-2) the complexation constant of cyclodextrin with phenolphthalein in various temperatures was estimated and found slightly higher than our results, although close enough. The Table below allows direct comparison.

This study

Literature data

T(oC)

Kc (×104 M-1)

Kc (×104 M-1)

20

2.23±0.08

3.94

30

1.34±0.06

2.20

40

0.81±0.05

1.20

A short discussion is on these issues is presented in the revised manuscript (lines:358-368).

Issue 2:

  1. 4 line 94: “In this study, water was the upper solvent and diethyl ether was the solvent at the bottom.” Likely the reverse is true. Please check.

Reply to Reviewer comment and changes made:

We agree with reviewer, and we revised this sentence. The correct sentence is: “In this study, diethyl ether was the upper solvent and water was the solvent at the bottom” (lines: 100-101).

Issue 3:

  1. 7. The trend reported in figure 3 are not consistent with those expected from Figure 2a and with what stated in line 226: “As cation size increases, the absorbance rises except for Li that has the smallest ionic radius.” Please check.

Reply to Reviewer comment and changes made:

Figure 2a merely shows the absorbance spectra of a single CD-PP solution after the addition of 100 mM of iodide salt corresponding to different alkali-type. Figure 3 represents the variation of the A0/A ratio as a function of β-cyclodextrin concentration after the addition of 100 mM of different alkali iodide salts. A is the absorbance of PP in the presence of CD, while A0 corresponds to the absorbance of pure PP (without CD). So, the curves in Figure 3 are determined by both A and A0. To make it clearer, the curve for Li+ lies in that position due to higher A0 than that of the K+.

Issue 4:

P.8 line 249: “The increase of the KI salt concentration induces a monotonous reduction of the KC values, which is expected since I anion is a competitor for the complexation” This can be also considered to explain the results mentioned above in line 235: “The results indicate that the complexation coefficient reduces after the addition of iodide salts.”, when different possibilities have been proposed (lines 237-242). Please clarify. I remark that if activity changes are considered, then equations (9) and (10) should be reformulated.

 Concerning the values reported in Table 2, I suggest expressing errors with only one meaningful digit; moreover the numbers of the related quantities must be expressed with an appropriate number of significant figures. In this respect it seems that the errors on DG are very high, also if compared to those on Kc. I remark that DG can be obtained directly from Kc, without the employ of eq. 13 that depends on DH and DS, which are, in turn, determined by the van’t Hoff plot.

Reply to Reviewer comment and changes made:

We agree with reviewer’s comment that I anion acts as competitor for the complexation, and this explains the behavior described in the manuscript. We added a short note on this issue in lines 242-244. Furthermore, we revised equation 13 according to reviewer’s suggestion (line 277).

The revised equation ( involves temperature and KC and has been used to recalculate the ΔG values that are presented in Table 1 and 2, respectively. The corresponding errors have been calculated utilizing the standard methodology of error propagation.

The activity change was not considered in this work due to its minor contribution. It may influence the interactions of PP-CD but not as strong as the formation of inclusion complexes of anions and cations with CD. All changes are highlighted with green color.

Issue 5:

  1. 9 line 282: “When ΔH > 0 and ΔS < 0, hydrophobic interactions dominate, while when ΔH < 0 and ΔS < 0, hydrogen bonding and van der Waals interactions play a major role. Electrostatic forces are considerable when the relations ΔH < 0 and ΔS > 0 hold [21].” This description is very qualitative and difficult to be fully understood. For example, hydrophobic interactions are typically entropically driven. Please check.

Reply to Reviewer comment and changes made:

A host-guest complex of CD has several forces which drive it. Van der Waals interactions, hydrogen bonding, hydrophobic interactions, electrostatic forces, and the release of water molecules from the cavity of the host to the bulk [doi.org/10.1021/cr960371r]. The value of the thermodynamic parameters is a contribution of the desolvation and the non-covalent interactions that we mentioned above. Generally, for the complexation of CDs the usual driving forces are hydrophobic interactions, van der Waals interactions and hydrogen bonding. Molecular docking results showed that van der Waals and hydrogen bonding predominate. When ΔH<0 and ΔS<0, the van der Waals interactions and hydrogen bonding are the driving forces for the complexation. The reason is that van der Waals interactions and hydrogen bonding between semi polar molecules is an exothermic process. Moreover, the conformational restriction and loss of rotational freedom during the formation of the inclusion complex yields unfavorable entropy loss [doi.org/10.1039/C9NR05790K]. When ΔΗ>0 and ΔS>0, the hydrophobic interactions are involved in the complexation. The hydrophobic interactions are entropy driven with a positive entropy change and a slightly positive enthalpy change. Lastly, when ΔΗ<0 and ΔS>0 the electrostatic forces play a significant role [doi.org/10.3390/molecules24244565].

These comments were added in the revised manuscript as a new paragraph and highlighted with yellow color. Three new references were also added, and the reference list is renumbered, accordingly (lines: 298-312).

Issue 6:

P.9 line 287: “The Gibbs free energy change overall reduces with increasing temperature indicating that the process becomes less spontaneous [22]” The process is in any case at equilibrium, is the position of the equilibrium that changes with temperature, as expected for an exothermic process.   

Reply to Reviewer comment and changes made:

The Gibbs free energy change overall reduces with increasing temperature indicating that the process becomes less spontaneous and shifts the equilibrium towards the reactants. This note is added in the revised manuscript (lines: 295-297).

Issue 7:

  1. 9 line 291: “With increasing the concentration of the KI salt, the absolute value of the enthalpy change decreases due to the competition between iodine anion of KI and phenolphthalein for the binding sites of β-cyclodextrin [19, 23].” Why does the competition is expected to reduce the enthalpy value?

Reply to Reviewer comment and changes made:

The formation of the inclusion complex is a two-step procedure. Firstly, water from the cavity of cyclodextrin is released to the bulk and then the guest molecule is inserted into the cavity. In the presence of a competitor anion in the solution, the guest molecule should replace both water and the relatively strong-attached anion to cyclodextrin. Also, the heat of the PP-CD formation is the sum of the enthalpy of the expelling anion from the CD cavity and the CD-PP binding enthalpy. Some of the heat from the formation of the inclusion complex of CD-PP is used for the expulsion of the anion and thus, reduces the enthalpy value of the complexation.

This comment is added in the revised manuscript (lines: 323-330). All changes are highlighted with green color.

Issue 8:

P.10 lines 307-326. This part is quite assertive invoking different mechanisms and possibilities, but it is not clear how the findings of the present work might clarify this complex situation. Moreover, since iodide is expected to compete with PP, it is crucial to know if the formation of cation-anion ion pairs can be neglected in the considered conditions.

Reply to Reviewer comment and changes made:

In the context of this work, we studied molecular docking possibilities all cations to PP molecule. The obtained results revealed the presence of a potential interaction, nevertheless with positive energy indicating that these interactions are very weak. We suggest that the Li+ interacts with PP making it harder for the latter to form van der Waals interaction or hydrogen bonds with CD. To our knowledge, the presence of a stable PP-cation pair is not reported in the literature.

Issue 9:

P.12 line 383: “Both spectra reveal a close resemblance at least in the number of peaks supporting the proposed complexation between CD and PP molecules.” It is very hard to understand how the comparison in Figure 7 might support the existence of the complex. Probably a comparison between the spectrum calculated for the free PP and that calculated for PP in the complex might be helpful to support the discussion reported in section 3.3, which in the present form is not very convincing.  

Reply to Reviewer comment and changes made:

The computed IR spectrum of the CD-PP complex presented in Figure 7 can be used as a yardstick to evaluate the reliability of the complex formation experimentally.

The “fingerprint” spectral region, from about 500 to 1500 cm-1, usually contains a very complicated series of absorptions. These are mainly due to all manner of bending vibrations within the molecule. It is much more difficult to pick out individual bonds in this region than it is in the “cleaner” region at higher wavenumbers. The importance of the fingerprint region is that each different compound produces a different pattern of troughs in this part of the spectrum. In our case, the pattern in the fingerprint region exhibits similarities in the number of peaks and therefore can be used to identify the complex. Almost all the main peaks observed experimentally for the complex are present in the theoretical spectrum of the complex implying that the structure of the formed compound is identical with the structure of the complex.

This procedure is a common practice in the literature concerning the formation of inclusion complexes. Please see for example the following references:

doi.org/10.3390/molecules26195881

doi.org/10.1016/j.molstruc.2010.08.008

doi.org/10.1016/j.vibspec.2013.09.006

Issue 10:

P.12 line 403: “Hydrogen bonding and Van der Waals interactions dominate the complexation process, while in the case of lithium electrostatic forces are also important.” This conclusion is not justified.

Reply to Reviewer comment and changes made:

As explained earlier, the sign of the ΔΗ and ΔS values allows us to elucidate in qualitative manner the type of the interactions that dominate in the complexation. The experimental results revealed that the CD-PP complexation process in the presence of Li+ exhibits a slightly negative ΔS value indicating that the electrostatic interactions play significant role in the complexation compared to the other cations. Nevertheless, since the value of ΔS is slightly negative, it seems that van der Waals and hydrogen bonding are still contributing to the formation of the complex.

This comment is added in the revised manuscript and is highlighted with green color (lines: 457-463).

Issue 11:

Table 1 is very large; it is probably better to include it as supplementary material.

Reply to Reviewer comment and changes made:

We agree with reviewer, and we moved Table 1 into Supporting Information. All Tables of the revised manuscript were renumbered, accordingly.

Issue 12:

  1. 4 line 87: “The layering technique, also known as the liquid diffusion technique, involves the slow diffusion of one solvent into another.” Probably one or more references should be cited here.

Reply to Reviewer comment and changes made:

We agree with reviewer’s comment and we added the necessary reference concerning the layering technique. The reference list is renumbered, accordingly.

Round 2

Reviewer 1 Report

-Use of semi-empirical methods (AM1 and PM3) for the estimation of the enthalpy change is an ancient practice. Also, I admit that the CD-PP complex is a fairly large system for quantum chemical calculations, but more advanced (better than B3LYP) DFT methods (such as wB97X-D or M06-2X) treating weak interactions are available now. I recommend the authors to use more refined calculations in future works.

-The authors denote the alkali metal cations as Li, Na, etc. at numerous places. These should all be corrected to Li+, Na+, etc.

-The experimental methods (UV-VIS, FT-IR) employed in the study may not be able to reveal the complexation of MI (M = Li, Na, Cs…) with CD in solution phase in molecular detail. That is why quantum chemical calculations for the structures of MI-CD and MI-PP-CD in solution would be important, without which the competition between MI and PP for complexing with CD in interpreting the changes in UV-VIS may only be a crude guess.

 My question is: Is the complexation of MI with CD only competitive with the formation of CD – PP complex? Can’t I- and PP both bind to CD to cause the observed changes in UV-VIS spectra?

 I recommend the authors to add a short discussion on this argument, providing the rationale for their interpretation.

-Some English expressions should be revised:

Line 250: reduces à decreases

Line 252-253: This behavior can be explained having in mind that inorganic salts à This behavior can be explained by considering that inorganic salts

Line 428: lye à lie

-The resolution of Figure 6 is not good. It should be improved.

-It is not clear from Figure 6 which of the three phenyl rings of PP is inside the hydrophobic CD cavity. The authors may add a short paragraph for this description.